# GLINKX: A SCALABLE UNIFIED FRAMEWORK FOR HOMOPHILOUS AND HETEROPHILOUS GRAPHS

## ABSTRACT

In graph learning, there have been two predominant inductive biases regarding graph-inspired architectures: On the one hand, higher-order interactions and message passing work well on homophilous graphs and are leveraged by GCNs and GATs. Such architectures, however, cannot easily scale to large real-world graphs. On the other hand, shallow (or node-level) models using ego features and adjacency embeddings work well in heterophilous graphs. In this work, we propose a novel scalable shallow method – GLINKX – that can work both on homophilous and heterophilous graphs. GLINKX leverages (i) novel monophilous label propagations (ii) ego/node features, (iii) knowledge graph embeddings as positional embeddings, (iv) node-level training, and (v) low-dimensional message passing. Formally, we prove novel error bounds and justify the components of GLINKX. Experimentally, we show its effectiveness on several homophilous and heterophilous datasets.

## 1 INTRODUCTION

In recent years, graph learning methods have emerged with a strong performance for various ML tasks. Graph ML methods leverage the topology of graphs underlying the data (Battaglia et al., 2018) to improve their performance. Two very important design options for proposing graph ML based architectures in the context of *node classification* are related to whether the data is *homophilous* or *heterophilous*.

For homophilous data – where neighboring nodes share similar labels (McPherson et al., 2001; Altenburger & Ugander, 2018a) – Graph Neural Network (GNN)-based methods are able to achieve high accuracy. Specifically, a broad subclass sucessfull GNNs are Graph Convolutional Networks (GCNs) (e.g., GCN, GAT, etc.) (Kipf & Welling, 2016; Veličković et al., 2017; Zhu et al., 2020). In the GCN paradigm, *message passing* and *higher-order interactions* help node classification tasks in the homophilous setting since such inductive biases tend to bring the learned representations of linked nodes close to each other. However, GCN-based architectures suffer from *scalability issues*. Performing (higher-order) propagations during the training stage are hard to scale in large graphs because the number of nodes grows exponentially with the increase of the filter receptive field. Thus, for practical purposes, GCN-based methods require *node sampling*, substantially increasing their training time. For this reason, architectures (Huang et al., 2020; Zhang et al., 2022b; Sun et al., 2021; Maurya et al., 2021; Rossi et al., 2020) that leverage propagations outside of the training loop (as a preprocessing step) have shown promising results in terms of scaling to large graphs.

In *heterophilous* datasets (Rogers et al., 2014), the nodes that are connected tend to have different labels. Currently, many works that address heterophily can be classified into two categories concerning scale. On the one hand, recent successful architectures (in terms of accuracy) (Jin et al., 2022a; Di Giovanni et al., 2022; Zheng et al., 2022b; Luan et al., 2021; Chien et al., 2020; Lei et al., 2022) that address heterophily resemble GCNs in terms of design and thus suffer from the same scalability issues. On the other hand, *shallow or node-level models* (see, e.g., (Lim et al., 2021; Zhong et al., 2022)), i.e., models that are treating graph data as tabular data and do not involve propagations during training, has shown a lot of promise for large heterophilous graphs. In (Lim et al., 2021), it is shown that combining ego embeddings (node features) and adjacency embeddings works in the heterophilous setting. One element that LINKX exploits via the adjacency embeddings is monophily (Altenburger & Ugander, 2018a;b), namely the similarity of the labels of a node's neighbors. However, their design is still impractical in real-world data since the method (LINKX) is *not* inductive (see Section 2),

and embedding the adjacency matrix directly requires many parameters in a model. In LINKX, the adjacency embedding of a node can alternatively be thought of as a *positional embedding* (PE) of the node in the graph, and recent developments (Kim et al., 2022; Dwivedi et al., 2021; Lim et al., 2021) have shown the importance of PEs in both homophilous and heterophilous settings. However, most of these works suggest PE parametrizations that are difficult to compute in large-scale settings. We argue that PEs can be obtained in a scalable manner by utilizing *knowledge graph embeddings*, which, according to prior work, can (El-Kishky et al., 2022; Lerer et al., 2019; Bordes et al., 2013; Yang et al., 2014) be trained in very large networks.

**Goal & Contribution:** In this work, we develop a scalable method for node classification that: (i) works both on homophilous and heterophilous graphs, (ii) is *simpler and faster* than conventional message passing networks (by avoiding the neighbor sampling and message passing overhead during training), and (iii) can work in both a *transductive* and an *inductive*[1] setting. For a method to be scalable, we argue that it should: (i) run models on node-scale (thus leveraging i.i.d. minibatching), (ii) avoid doing message passing during training and do it a constant number of times before training, and (iii) transmit small messages along the edges. Our proposed method – GLINKX (see Section 3) – combines all the above desiderata. GLINKX has three components: (i) ego embeddings [2], (ii) PEs inspired by architectures suited for heterophilous settings, and (iii) *scalable* 2nd-hop-neighborhood propagations inspired by architectures suited for monophilous settings (Section 2.5). We provide novel theoretical error bounds and justify components of our method (Section 3.4). Finally, we evaluate GLINKX's empirical effectiveness on several homophilous and heterophilous datasets (Section 4).

## 2 PRELIMINARIES

### 2.1 NOTATION

We denote scalars with lower-case, vectors with bold lower-case letters, and matrices with bold upper-case. We consider a directed graph $G = G(V, E)$ with vertex set $V$ with $|V| = n$ nodes, and edge set $E$ with $|E| = m$ edges, and adjacency matrix $\boldsymbol{A}$. $\boldsymbol{X} \in \mathbb{R}^{n \times d_X}$ represents the $d_X$-dimensional features and $\boldsymbol{P} \in \mathbb{R}^{n \times d_P}$ represent the $d_P$-dimensional PE matrix. A node $i$ has a feature vector $\boldsymbol{x}_i \in \mathbb{R}^{d_x}$ and a positional embedding $\boldsymbol{p}_i \in \mathbb{R}^{d_P}$ and belongs to a class $y_i \in \{1, \ldots, c\}$. The training set is denoted by $G_{\text{train}}(V_{\text{train}}, E_{\text{train}})$, the validation set by $G_{\text{valid}}(V_{\text{valid}}, E_{\text{valid}})$, and test set by $G_{\text{test}}(V_{\text{test}}, E_{\text{test}})$. $\mathbb{I}\{\cdot\}$ is the indicator function. $\Gamma_c$ is the $c$-dimensional simplex.

### 2.2 GRAPH CONVOLUTIONAL NEURAL NETWORKS

In homophilous datasets, GCN-based methods have been used for node classification. GCNs (Kipf & Welling, 2016) utilize feature propagations together with non-linearities to produce node embeddings. Specifically, a GCN consists of multiple layers where each layer $i$ collects $i$-th hop information from the nodes through propagations and forwards this information to the $i + 1$-th layer. More specifically, if $G$ has a symmetrically-normalized adjacency matrix $\boldsymbol{A}'_{sym}$ (with self-loops) (ignoring the directionality of edges), then a GCN layer has the form

$$\boldsymbol{H}^{(0)} = \boldsymbol{X}, \ \boldsymbol{H}^{(i+1)} = \sigma\left(\boldsymbol{A}'_{sym}\boldsymbol{H}^{(i)}\boldsymbol{W}^{(i)}\right) \forall i \in [L], \ \boldsymbol{Y} = \text{softmax}\left(\boldsymbol{H}^{(L)}\right).$$

Here $\boldsymbol{H}^{(i)}$ is the embedding from the previous layer, $\boldsymbol{W}^{(i)}$ is a learnable projection matrix and $\sigma(\cdot)$ is a non-linearity (e.g. ReLU, sigmoid, etc.).

### 2.3 LINKX

In heterophilous datasets, the simple method of LINKX has been shown to perform well. LINKX combines two components – MLP on the node features $\boldsymbol{X}$ and LINK regression (Altenburger & Ugander, 2018a) on the adjacency matrix – as follows:

---

[1]For this paper, we operate in the *transductive setting*. See App. B for the inductive setting.
[2]We use ego embeddings and node features interchangeably.

---

**Algorithm 1** GLINKX Algorithm

---

**Input:** Graph $G(V, E)$ with train set $V_{\text{train}} \subseteq V$, node features $\boldsymbol{X}$, labels $\boldsymbol{Y}$
**Output:** Node Label Predictions $\boldsymbol{Y}_{\text{final}}$
**1st Stage (KGEs).** Pre-train knowledge graph embeddings $\boldsymbol{P}$ with Pytorch Biggraph.
**2nd Stage (MLaP).** Propagate labels and predict neighbor distribution
    1. **MLaP Forward:** Calculate the distribution of each training node's neighbors, i.e.
        $\hat{\boldsymbol{y}}_i = \frac{\sum_{j \in V_{\text{train}}:(j,i) \in E_{\text{train}}} \boldsymbol{y}_j}{|\{j \in V_{\text{train}}:(j,i) \in E_{\text{train}}\}|}$ for all $i \in V_{\text{train}}$
    2. **Learn distribution of a node's neighbors:**
        (a) For each epoch, calculate $\tilde{\boldsymbol{y}}_i = f_1(\boldsymbol{x}_i, \boldsymbol{p}_i; \boldsymbol{\theta}_1)$ for $i \in V_{\text{train}}$
        (b) Update the parameters s.t. the negative cross-entropy
            $\mathcal{L}_{\text{CE},1}(\boldsymbol{\theta}_1) = \sum_{i \in V_{\text{train}}} \text{CE}(\hat{\boldsymbol{y}}_i, \tilde{\boldsymbol{y}}_i; \boldsymbol{\theta}_1)$ is maximized in order to bring $\tilde{\boldsymbol{y}}_i$ statistically close to $\hat{\boldsymbol{y}}_i$.
        (c) Let $\boldsymbol{\theta}_1^*$ be the parameters at the end of the training that correspond to the epoch with the best validation accuracy.
    3. **MLaP Backward:** Calculate $\boldsymbol{y}_i' = \frac{\sum_{j \in V:(i,j) \in E} \tilde{\boldsymbol{y}}_j}{|\{j \in V:(i,j) \in E\}|}$ for all $i \in V_{\text{train}}$, where
        $\tilde{\boldsymbol{y}}_j = f_1(\boldsymbol{x}_j, \boldsymbol{p}_j; \boldsymbol{\theta}_1^*)$.
**3rd Stage (Final Model).** Predicting node's own label distribution:
    1. For each epoch, calculate $y_{\text{final},i} = f_2(\boldsymbol{x}_i, \boldsymbol{p}_i, \boldsymbol{y}_i'; \boldsymbol{\theta}_2)$.
    2. Update the parameters s.t. the negative cross-entropy
        $\mathcal{L}_{\text{CE},2}(\boldsymbol{\theta}_2) = \sum_{i \in V_{\text{train}}} \text{CE}(y_i, y_{\text{final},i}; \boldsymbol{\theta}_2)$ is maximized.
Return $\boldsymbol{Y}_{\text{final}}$

---

$$\boldsymbol{H}_X = \text{MLP}_X(\boldsymbol{X}), \ \boldsymbol{H}_A = \text{MLP}_A(\boldsymbol{A}), \ \boldsymbol{Y} = \text{ResNet}(\boldsymbol{H}_X, \boldsymbol{H}_A).$$

## 2.4 Node Classification

In node classification problems on graphs, we have a model $f(\boldsymbol{X}, \boldsymbol{Y}_{\text{train}}, \boldsymbol{A}; \boldsymbol{\theta})$ that takes as an input the node features $\boldsymbol{X}$, the training labels $\boldsymbol{Y}_{\text{train}}$ and the graph topology $\boldsymbol{A}$ and produces a prediction for each node $i$ of $G$, which corresponds to the probability that a given node belongs to any of $c$ classes (with the sum of such probabilities being one). The model is trained with back-propagation. Once trained, the model can be used for the prediction of labels of nodes in the test set.

There are two training regimes: *transductive* and *inductive*. In the transductive training regime, we have full knowledge of the graph topology (for the train, test, and validation sets) and the node features, and the task is to predict the labels of the validation and test set. In the inductive regime, only the graph induced by $V_{\text{train}}$ is known at the time of training, and then the full graph is revealed for prediction on the validation and test sets. In real-world scenarios, such as online social networks, the dynamic nature of problems makes the inductive regime particularly useful.

## 2.5 Homophily, Heterophily & Monophily

***Homophily and Heterophily:*** There are various measures of homophily in the GNN literature like node homophily and edge homophily Lim et al. (2021). Intuitively, homophily in a graph implies that nodes with similar labels are connected. GNN-based approaches like GCN, GAT, etc., leverage this property to improve the node classification performance. Alternatively, if a graph has low homophily – namely, nodes that connect tend to have different labels – it is said to be *heterophilous*. In other words, a graph is heterophilous if neighboring nodes do not share similar labels.

***Monophily:*** Generally, we define a graph to be monophilous if the label of a node is similar to that of its neighbors' neighbors[3]. Etymologically, the word "monophily" is derived from the Greek words *"monos"* (unique) and *"philos"* (friend), which in our context means that a node – regardless of its label – has neighbors of primarily one label. In the context of a directed graph, monophily can be

---

[3]A similar definition of monophily has appeared in (Altenburger & Ugander, 2018a), whereby many nodes have extreme preferences for connecting to a certain class.

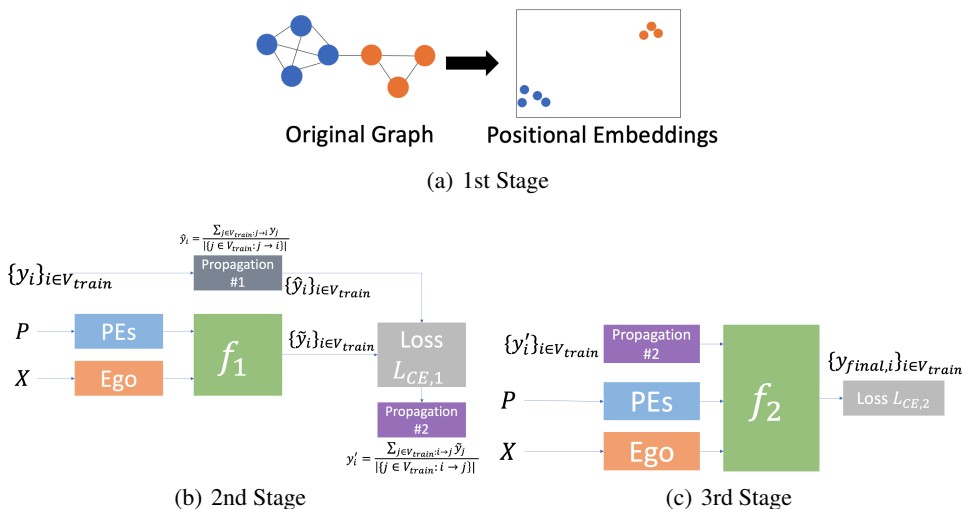

Figure 1: Block Diagrams of GLINKX stages.

thought of as a structure that resembles Fig. 2(a) where similar nodes (in this case, three green nodes connected to a yellow node) are connected to a node with a different label.

We argue that encoding monophily into a model **can be helpful for both heterophilous and homophilous graphs (see Figs. 3(b) and 3(c)), which is one of the main motivators behind our work**. In homophilous graphs, monophily will fundamentally encode the 2nd-hop neighbor's label information, and since in such graphs, neighboring nodes have similar labels, it can provide a helpful signal for node classification. In heterophily, neighboring nodes have different labels, but the 2nd-hop neighbors may share the same label, providing helpful information for node classification. Monophily is effective for heterophilous graphs (Lim et al., 2021). Therefore, an approach encoding monophily has an advantage over methods designed specifically for homophilous and heterophilous graphs, especially when varying levels of homophily can exist between different sub-regions in the same graph (see Section 3.3). It may also not be apparent if the (sub-)graph is purely homophilous/heterophilous (since these are not binary constructs), which makes a unified architecture that can leverage graph information for both settings all the more important.

## 3 METHOD

### 3.1 COMPONENTS & MOTIVATION

The desiderata we laid down on Section 1 can be realized by three components: (i) PEs, (ii) ego embeddings, and (iii) label propagations that encode monophily. More specifically, ego embeddings and PEs are used as primary features, which have been shown to work for both homophilous and heterophilous graphs for the models we end up training. Finally, the propagation step is used to encode monophily to provide additional information to our final prediction.

***Positional Embeddings:*** We use PEs to provide our model information about the position of each node and hypothesize that PEs are an important piece of information in the context of large-scale node classification. PEs have been used to help discriminate isomorphic graph (sub)-structures (Kim et al., 2022; Dwivedi et al., 2021; Srinivasan & Ribeiro, 2019). This is useful for both homophily (Kim et al., 2022; Dwivedi et al., 2021) and heterophily (Lim et al., 2021) because isomorphic (sub)-structures can exist in both the settings. In the homophilous case, adding positional information can help distinguish nodes that have the same neighborhood but distinct position (Dwivedi et al., 2021; Morris et al., 2019; Xu et al., 2019), circumventing the need to do higher-order propagations (Dwivedi et al., 2021; Li et al., 2019; Bresson & Laurent, 2017) which are prone to over-squashing (Alon & Yahav, 2021). In heterophily, structural similarity among nodes is important for classification, as in the case of LINKX – where adjacency embedding can be considered a PE. However, in large graphs, using

adjacency embeddings or Laplacian eigenvectors (as methods such as (Kim et al., 2022) suggest) can be a computational bottleneck and may be infeasible.

In this work, we leverage *knowledge graph embeddings* (KGEs) to encode positional information about the nodes, and embed the graph. Using KGEs has two benefits: Firstly, KGEs can be trained quickly for large graphs. This is because KGEs compress the adjacency matrix into a fixed-sized embedding, and adjacency matrices have been shown to be effective in heterophilous cases. Further, KGEs are lower-dimensional than the adjacency matrix (e.g., $d_P \sim 10^2$), allowing for faster training and inference times. Secondly, KGEs can be pre-trained efficiently on such graphs (Lerer et al., 2019) and can be used off-the-shelf for other downstream tasks, including node classification (El-Kishky et al., 2022)[4]. So, in the 1st Stage of our methods in Alg. 1 (Fig. 1(a)) we train KGEs model on the available graph structure. Here, we fix this positional encoding once they are pre-trained for downstream usage. Finally, we note that this step is transductive but we can easily make it inductive (El-Kishky et al., 2022; Albooyeh et al., 2020).

***Ego Embeddings:*** We get ego embeddings from the node features. Such embeddings have been used in homophilous and heterophilous settings (Lim et al., 2021; Zhu et al., 2020). Node embeddings are useful for tasks where the graph structure provides little/no information about the task.

***Monophilous Label Propagations:*** We now propose a novel monophily (refer Section 2.5) inspired label propagation which we refer to as Monophilous Label Propagation (MLaP). MLaP has the advantage that we can use it both for homophilous and heterophilous graphs or in a scenario with varying levels of graph homophily (see Section 3.3) as it encodes monophily (Section 2.5).

To understand how MLaP encodes monophily, we consider the example in Fig. 2. In this example, we have three green nodes connected to a yellow node and two nodes of different colors connected to the yellow node. Then, one way to encode monophily in Fig. 2(a) while predicting label for $j_\ell, \ell \in [5]$, is to get a *distribution* of labels of nodes connected to node $i$ thus encoding its neighbors' distribution. The fact that there are more nodes with green color than other colors can be used by the model to make a prediction. But this information may only sometimes be present, or there may be few labeled nodes around node $i$. Consequently, we propose to use a model that predicts the label distribution of nodes connected to $i$. We use the node features ($\boldsymbol{x}_i$) and PE ($\boldsymbol{p}_i$) of node $i$ to build this model since nodes that are connected to node $i$ share similar labels, and thus, the features of node $i$ must be predictive of its neighbors. So, in Fig. 2(a), we train a model to predict a distribution of $i$'s neighbors. Next, we provide $j_\ell$ the learned distribution of $i$'s neighbors by propagating the learned distribution from $i$ back to $j_\ell$. Eqs. (1) to (3) correspond to MLaP. We train a final model that leverages this information together with node features and PEs (Fig. 2(b)).

## 3.2 OUR METHOD: GLINKX

We put the components discussed in Section 3.1 together into three stages. In the first stage, we pre-train the PEs by using KGEs. Next, encode monophily into our model by training a model that predicts a node's neighbors' distribution and by propagating the soft labels from the fitted model. Finally, we combine the propagated information, node features, and PEs to train a final model. GLINKX is described in Alg. 1 and consists of three main components detailed as block diagrams in Fig. 1. Fig. 2 shows the GLINKX stages from Alg. 1 on a toy graph:

*1st Stage (KGEs):* We train DistMult KGEs with Pytorch-Biggraph (Yang et al., 2014) treating $G$ as a knowledge graph with only one relation (see App. A.4 for more details). Here we have decided to use DistMult, but one can use their method of choice to embed the graph.

*2nd Stage (MLaP):* First (2nd Stage in Alg. 1, Fig. 1(b), and Fig. 2(a)), for a node we want to learn the distribution of *its neighbors*. To achieve this, we propagate the labels from a node's neighbors (we call this step MLaP Forward), i.e. calculate

$$\hat{\boldsymbol{y}}_i = \frac{\sum_{j \in V_{\text{train}}:(j,i) \in E_{\text{train}}} \boldsymbol{y}_j}{|\{j \in V_{\text{train}} : (j,i) \in E_{\text{train}}\}|} \qquad \forall i \in V_{\text{train}}. \tag{1}$$

---

[4]Positional information can also be provided by other methods such as node2vec (Grover & Leskovec, 2016), however, most of such methods are less scalable.

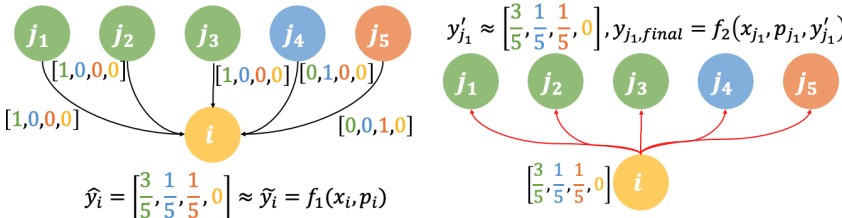

(a) MLaP Forward & Neighbor Model      (b) MLaP Backward & Final Model

Figure 2: Example. For node $i$ we want to learn a model that takes $i$'s features $\boldsymbol{x}_i \in \mathbb{R}^{d_X}$, and PEs $\boldsymbol{p}_i \in \mathbb{R}^{d_P}$ and predict a value $\widetilde{\boldsymbol{y}}_i \in \mathbb{R}^c$ that matches the label distribution of it's neighbors neighbors $\hat{\boldsymbol{y}}_i$ using a shallow model. Next, we want to propagate (outside the training loop) the (predicted) distribution of a node back to its neighbors and use it together with the ego features and the PEs to make a prediction about a node's own label. We propagate $\tilde{\boldsymbol{y}}_i$ to its neighbors $j_1$ to $j_5$. For example, for $j_1$, we encode the propagated distribution estimate $\tilde{\boldsymbol{y}}_i$ from $i$ to form $\boldsymbol{y}'_{j_1}$. We predict the label by using $\boldsymbol{y}'_{j_1}, \boldsymbol{x}_{j_1}, \boldsymbol{p}_{j_1}$.

Then, we train a model that predicts the distribution of neighbors, which we denote with $\tilde{\boldsymbol{y}}_i$ using the ego features $\{\boldsymbol{x}_i\}_{i \in V_{\text{train}}}$ and the PEs $\{\boldsymbol{p}_i\}_{i \in V_{\text{train}}}$ and maximize the negative cross-entropy with treating $\{\hat{\boldsymbol{y}}_i\}_{i \in V_{\text{train}}}$ as ground truth labels, namely we maximize

$$\mathcal{L}_{\text{CE, 1}}(\boldsymbol{\theta}_1) = \sum_{i \in V_{\text{train}}} \sum_{l \in [c]} \hat{\boldsymbol{y}}_{i,l} \log(\tilde{\boldsymbol{y}}_{i,l}), \tag{2}$$

where $\tilde{\boldsymbol{y}}_i = f_1(\boldsymbol{x}_i, \boldsymbol{p}_i; \boldsymbol{\theta}_1)$ and $\boldsymbol{\theta}_1 \in \Theta_1$ is a learnable parameter vector. Although in this paper we assume to be in the *transductive setting*, this step allows us to be inductive (see App. B). In Section 3.4 we give a theoretical justification of this step, namely *"why is it good to use a parametric model to predict the distribution of neighbors?"*.

Finally, we propagate the predicted soft-labels $\tilde{\boldsymbol{y}}_i$ back to the original nodes, i.e. calculate

$$\boldsymbol{y}'_i = \frac{\sum_{j \in V: (i,j) \in E} \tilde{\boldsymbol{y}}_j}{|\{j \in V : (i,j) \in E\}|} \quad \forall i \in V_{\text{train}}, \tag{3}$$

where the soft labels $\{\tilde{\boldsymbol{y}}_i\}_{i \in V_{\text{train}}}$ have been computed with the parameter $\boldsymbol{\theta}_1^*$ of the epoch with the best validation accuracy from model $f_1(\cdot | \boldsymbol{\theta}_1)$. We call this step MLaP Backward.

*3rd Stage (Final Model):* We make the final predictions $\boldsymbol{y}_{\text{final, i}} = f_2(\boldsymbol{x}_i, \boldsymbol{p}_i, \boldsymbol{y}'_i; \boldsymbol{\theta}_2)$ by combining the ego embeddings, PEs, and the (back)-propagated soft labels ($\boldsymbol{\theta}_2$ is a learnable parameter vector). We use the soft-labels $\tilde{\boldsymbol{y}}_i$ instead of the actual labels one-hot ($y_i$) in order to avoid label leakage, which hurts performance (see also (Shi et al., 2020) for a different way to combat label leakage). Finally, we maximize the negative cross-entropy with respect to a node's own labels,

$$\mathcal{L}_{\text{CE, 2}}(\boldsymbol{\theta}_2) = \sum_{i \in V_{\text{train}}} \sum_{l \in [c]} \mathbb{I}\{y_i = l\} \log(\boldsymbol{y}_{\text{final, i},l}), \tag{4}$$

Overall, Stage 2 corresponds to learning the neighbor distributions and propagating these distributions, and Stage 3 uses these distributions to train a new model which predicts a node's labels. In Section 3.4, we prove that such a two-step procedure incurs lower errors than directly using the features to predict a node's labels.

*Scalability:* GLINKX is highly scalable as it performs message passing a constant number of times by paying an $O(mc)$ cost, where the dimensionality of classes $c$ is usually small (compared to $d_X$ that GCNs rely on). In both Stages 2 and 3 of Alg. 1, we train node-level MLPs, which allow us to leverage i.i.d. (row-wise) mini-batching, like tabular data, and thus our complexity is similar to other shallow methods (LINKX, FSGNN) (Lim et al., 2021; Maurya et al., 2021). This, combined with the

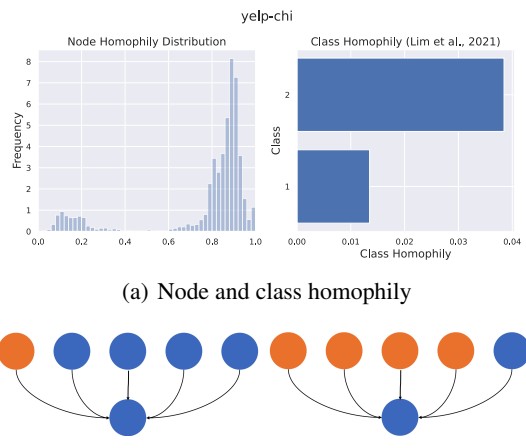

(a) Node and class homophily

(b) Homophilous example  (c) Heterophilous example

Figure 3: Top: Node and class homophily distributions for the yelp-chi dataset. Bottom: Examples of a homophilous (Fig. 3(b)) and a heterophilous (Fig. 3(c)) region in the same graph that are both monophilous, namely they are connected to many neighbors of the same kind. In a spam network, the homophilous region corresponds to many non-spam reviews connecting to non-spam reviews (which is the expected behaviour of a non-spammer user), and the heterophilous region corresponds to spam reviews targeting non-spam reviews (which is the expected behaviour of spammers), thus, yielding a graph with both homophilous and heterophilous regions such as in Fig. 3(a).

propagation outside the training loops, circumvents the scalability issues of GCNs. For more details, refer App. A.3.

## 3.3 VARYING HOMOPHILY

Graphs with monophily experience homophily, heterophily, or both. For instance, in the yelp-chi dataset – where we classify a review as spam/non-spam (see Fig. 3) – we observe a case of monophily together with varying homophily. Specifically in this dataset, spam reviews are linked to non-spam reviews, and non-spam reviews usually connect to other non-spam reviews, which makes the node homophily distribution bimodal. Here the 2nd-order similarity makes the MLaP mechanism effective.

## 3.4 THEORETICAL ANALYSIS

***Justification of Stage 2:*** In Stage 2, we train a parametric model to learn the distribution of a node's neighbors from the node features $\boldsymbol{\xi}_i$[5]. Arguably, we can learn such a distribution naïvely by counting the neighbors $i$ that belong to each class. This motivates our first theoretical result. In summary, we show that training a parametric model for learning the distribution of a node's neighbors (as in Stage 2) yields a lower error than the naïve solution. Below we present the Thm. 1 (proof in App. F) for undirected graphs (the case of directed graphs is the same, but we omit it for simplicity of exposition):

**Theorem 1.** *Let $G([n], E)$ be an undirected graph of minimum degree $K > c^2$ and let $\boldsymbol{Q}_i \in \Gamma_c$ be the likelihood, from the viewpoint of node $i$, of any node in its neighborhood $\mathcal{N}(i)$ to be assigned to different classes for every node $i \in [n]$. The following two facts are true (under standard assumptions for SGD and the losses):*

1. *Let $\widehat{\boldsymbol{Q}}_i$ be the sample average of $\boldsymbol{Q}_i$, i.e. $\widehat{Q}_{i,j} = \frac{1}{|\mathcal{N}(i)|} \sum_{k \in \mathcal{N}(i)} \mathbb{I}\{y_k = j\}$. Then, for every $i \in [n]$, we have that $\max_{j \in [c]} \mathbb{E}[|Q_{i,j} - \widehat{Q}_{i,j}|] \leq \mathbb{E}[\|\boldsymbol{Q}_i - \widehat{\boldsymbol{Q}}_i\|_\infty] \leq O\left(\sqrt{\frac{\log(Kc)}{K}}\right)$.*

2. *Let $q(\cdot|\boldsymbol{\xi}_i; \boldsymbol{\theta})$ be a model parametrized by $\boldsymbol{\theta} \in \mathbb{R}^D$ that uses the features $\boldsymbol{\xi}_i$ of each node $i$ to predict $\boldsymbol{Q}_i$. We estimate the parameter $\boldsymbol{\theta}_1$ by running SGD for $t = n$ steps to maximize $\mathcal{L}(\boldsymbol{\theta}) = \frac{1}{n} \sum_{i=1}^n \sum_{j=1}^c Q_{i,j} \log q(j|\boldsymbol{\xi}_i; \boldsymbol{\theta})$. Then, for every $i \in [n]$, we have that $\max_{j \in [c]} \mathbb{E}[|q(j; \boldsymbol{\xi}_i; \boldsymbol{\theta}_1) - Q_{i,j}|] \leq O\left(\sqrt{\frac{\log n}{n}}\right)$.*

---

[5]In Section 3.1, $\boldsymbol{\xi}_i$s correspond to the augmented features $\boldsymbol{\xi}_i = [\boldsymbol{x}_i; \boldsymbol{p}_i]$

It is evident here that if the minimum degree $K$ is much smaller than $n$, then the parametric model has lower error than the naïve approach, namely $\tilde{O}(n^{-1/2})$ compared to $\tilde{O}(K^{-1/2})$.

***Justification of Stages 2 and 3:*** We now provide theoretical foundations for the two-stage approach. Specifically, we argue that a two-stage procedure involving learning the distribution of a node's 2nd-hop neighbor distributions (we assume for simplicity, again, that the graph is undirected) first with a parametric model such as in Thm. 1, and then running a two-phase algorithm to learn a parametric model that predicts a node's label, yields a lower error than naïvely training a shallow parametric model to learn a node's labels. The first phase of the two-phase algorithm involves training the model first by minimizing the cross-entropy between the predictions and the 2nd-hop neighborhood distribution. Then the model trains a joint objective that uses the learned neighbor distributions and the actual labels starting from the model learned in the previous phase.

**Theorem 2.** *Let $G([n], E)$ be an undirected graph of minimum degree $K > c^2$ and, let $P_i$ be the likelihood of node $i$ to be assigned to a different class, and let $Q_i, q(\cdot|\xi_i; \theta_1)$ defined as in Thm. 1. Let $p(\cdot|\xi_i; w)$ be a model parametrized by $w \in \mathbb{R}^D$ that is used to predict the class assignments $y_i \sim p(\cdot|\xi_i; w)$. Let $w_*$ be the optimal parameter. The following are true (under standard assumptions for SGD and the losses):*

1. *The naïve optimization scheme that runs SGD to maximize $\mathcal{G}(w) = \frac{1}{n} \sum_{i=1}^{n} \sum_{j=1}^{c} P_{i,j} \log p(j|\xi_i; w)$ for $n$ steps has error $\mathbb{E}[\mathcal{G}(w_{n+1}) - \mathcal{G}(w_*)] \leq O\left(\frac{\log n}{n}\right)$.*

2. *The two-phase optimization scheme that runs SGD to maximize $\widehat{\mathcal{G}}(w) = \frac{1}{n} \sum_{i=1}^{n} \sum_{j=1}^{c} \left(\frac{1}{|\mathcal{N}(i)|} \sum_{k \in \mathcal{N}(i)} q(j|\xi_k; \theta_1)\right) \log p(j|\xi_i; w)$ for $n_1$ steps, to estimate a solution $w'$ and then runs SGD on the objective $\lambda \widehat{\mathcal{G}}(w) + (1 - \lambda)\mathcal{G}(w)$ for $n$ steps starting from $w'$, achieves error $\mathbb{E}[\mathcal{G}(w_{n+1}) - \mathcal{G}(w_*)] \leq O\left(\frac{\sqrt{\log n \log \log n}}{n}\right)$.*

You can find the proof in App. F. We observe that the two-phase optimization scheme can reduce the error by a factor of $\sqrt{\log n / \log \log n}$ highlighting the importance of using the distribution of the 2nd-hop neighbors of a node to predict its label and holds regardless of the homophily properties of the graph. Also, note that the above two-phase optimization scheme differs from the description of the method we gave in Alg. 1. The difference is that the distribution of a node's neighbors is embedded into the model in the case of Alg. 1, and the distribution of a node's neighbors are embedded into the loss function in Thm. 2 as a regularizer. In Alg. 1, we chose to incorporate this information in the model because using multiple losses harms scalability and makes training harder in practice. In the same spirit, the conception of GCNs (Kipf & Welling, 2016) replaces explicit regularization with the graph Laplacian with the topology into the model (see also (Hamilton et al., 2017; Yang et al., 2016)).

## 3.5 COMPLEMENTARITY

Different components of GLINKX provide a *complementary* signal to components proposed in the GNN literature (Maurya et al., 2021; Zhang et al., 2022b; Rossi et al., 2020). One can combine GLINKX with existing architectures (e.g. feature propagations (Maurya et al., 2021; Rossi et al., 2020), label propagations (Zhang et al., 2022b)) for potential metric gains. For example, SIGN computes a series of $r \in \mathbb{N}$ feature propagations $[X, \Phi X, \Phi^2 X, \ldots, \Phi^r X]$ where $\Phi$ is a matrix (e.g., normalized adjacency or normalized Laplacian) as a preprocessing step. We can include this complementary signal, namely, embed each of the propagated features and combine them in the 3rd Stage to GLINKX. Overall, although in this paper we want to keep GLINKX simple to highlight its main components, we conjecture that adding more components to GLINKX would improve its performance on datasets with highly variable homophily (see Section 3.3).

## 4 EXPERIMENTS & CONCLUSIONS

***Comparisons:*** We experiment with homophilous and heterophilous datasets (see Tab. 1 and App. D.3). We train KGEs with Pytorch-Biggraph (Lerer et al., 2019; Yang et al., 2014). For homophilous datasets, we compare with vanilla GCN and GAT, FSGNN, and Label Propagation (LP). For a fair comparison, we compare with one-layer GCN/GAT/FSGNN/LP since our method is one-hop. We also compare with higher-order (h.o.) GCN/GAT/FSGNN/LP with 2 and 3 layers. In the heterophilous

Table 1: Experimental results. (*) = results from the OGB leaderboard.

| | Homophilous Datasets | | Heterophilous Datasets | | |
|---|---|---|---|---|---|
| | PubMed | ogbn-arxiv | squirrel | yelp-chi | arxiv-year |
| $n$ | 19.7K | 169.3K | 5.2K | 169.3K | 45.9K |
| $m$ | 44.3K | 1.16M | 216.9K | 7.73M | 1.16M |
| Edge-insensitive homophily (Lim et al., 2021) | 0.66 | 0.41 | 0.02 | 0.05 | 0.27 |
| $d_X / c$ | 500 / 27 | 128 / 40 | 2089 / 5 | 32 / 2 | 128 / 5 |
| GLINKX w/ KGEs | 87.95 ±0.30 | **69.27 ±0.25** | 45.83 ±2.89 | 87.82 ±0.20 | **54.09 ±0.61** |
| GLINKX w/ Adjacency | **88.03 ±0.30** | 69.09 ±0.13 | **69.15 ±1.87** | **89.32 ±0.45** | 53.07 ±0.29 |
| Label Propagation (1-hop) | 83.02 ±0.35 | **69.59 ±0.00** | 32.22 ±1.45 | 85.98 ±0.28 | 43.71 ±0.22 |
| LINKX (from (Lim et al., 2021)) | 87.86 ±0.77 | 67.32 ±0.24 | 61.81 ±1.80 | 85.86 ±0.40 | **56.00 ±1.34** |
| LINKX (our runs) | 87.55 ±0.37 | 63.91 ±0.18 | 61.46 ±1.60 | 88.25 ±0.24 | 53.78 ±0.06 |
| GCN w/ 1 Layer | 86.43 ±0.74 | 50.76 ±0.20 | | N/A | |
| GAT w/ 1 Layer | 86.41 ±0.53 | 54.42 ±0.10 | | N/A | |
| FSGNN w/ 1 Layer | 88.93 ±0.31 | 61.82 ±0.84 | 64.06 ±2.69 | 86.36 ±0.36 | 42.86 ±0.22 |
| Higher-order GCN | 86.29 ±0.46 | 71.18 ±0.27 (*) | | N/A | |
| Higher-order GAT | 86.64 ±0.40 | **73.66 ±0.11** (*) | | N/A | |
| Higher-order FSGNN | **89.37 ±0.49** | 69.26 ±0.36 | 68.04 ±2.19 | 86.33 ±0.30 | 44.89 ±0.29 |
| Label Propagation (2-hop) | 83.44 ±0.35 | 69.78 ±0.00 | 43.41 ±1.44 | 85.95 ±0.26 | 46.30 ±0.27 |
| Label Prop. on $\mathbb{I}[A^2 - A - I \geq 0]$ | 82.14 ±0.33 | 9.87 ±0.00 | 24.43 ±1.18 | 85.68 ±0.32 | 23.08 ±0.13 |

Table 2: Ablation Study. We use the hyperparameters of the best run from Tab. 1 with KGEs.

| | Ablation Type | Stages | All | Remove ego embeddings | Remove propagation | Remove PEs |
|---|---|---|---|---|---|---|
| Heterophilous | arxiv-year | All Stages | 54.09 ±0.61 | 53.52 ±0.77 | 50.83 ±0.24 | 39.06 ±0.35 |
| | arxiv-year | 3rd Stage | 54.09 ±0.61 | 53.69 ±0.65 | 50.83 ±0.24 | 49.13 ±1.10 |
| Homophilous | ogbn-arxiv | All Stages | 69.27 ±0.25 | 61.26 ±0.33 | 62.70 ±0.34 | 65.64 ±0.18 |
| | ogbn-arxiv | 3rd Stage | 69.27 ±0.25 | 67.60 ±0.39 | 62.70 ±0.34 | 69.62 ±0.15 |

case, we compare with LINKX[6] because it is scalable and is shown to work better than other baselines (e.g., H2GCN, MixHop, etc.) and with FSGNN. Note that we do not compare GLINKX with other more complex methods because GLINKX is complementary to them (see Section 3.5), and we can incorporate these designs into GLINKX. We use a *ResNet* module to combine our algorithm's components from Stages 2 and 3. Details about the hyperparameters we use are in App. C.

In the heterophilous datasets, GLINKX outperforms LINKX (except for arxiv-year, where we are within the confidence interval). Moreover, the performance gap between using KGEs and adjacency embeddings shrinks as the dataset grows. In the homophilous datasets, GLINKX outperforms 1-layer GCN/GAT/LP/FSGNN and LINKX. In PubMed, GLINKX beats h.o. GCN/GAT and in arxiv-year GLINKX is very close to the performance of GCN/GAT.

Finally, we note that our method produces consistent results across regime shifts. In detail, in the heterophilous regime, our method performs on par with LINKX; however, when we shift to the homophilous regime, LINKX's performance drops, whereas our method's performance continues to be high. Similarly, while FSGNN performs similarly to GLINKX on the homophilous datasets, we observe a significant performance drop on the heterophilous datasets (see arxiv-year).

***Ablation Study:*** We ablate each component of Alg. 1 to see each component's performance contribution. We use the hyperparameters of the best model from Tab. 1. We perform two types of ablations: (i) we remove each of the components from all stages of the training, and (ii) we remove the corresponding components only from the 3rd Stage. Except for removing the PEs from the 3rd Stage only on ogbn-arxiv, all components contribute to increased performance on both datasets. Note that adding PEs in the 1st Stage does improve performance, suggesting the primary use case of PEs.

***Conclusion:*** We present GLINKX, a scalable method for node classification in homophilous and heterophilous graphs that combines three components: (i) ego embeddings, (ii) PEs, and (iii) monophilous propagations. As future work, (i) GLINKX can be extended in heterogeneous graphs, (ii) use more expressive methods such as attention or Wasserstein barycenters (Cuturi & Doucet, 2014) for averaging the low-dimensional messages, and (iii) add complementary signals.

---

[6]We have run our method with hyperparameter space that is a subset of the sweeps reported in (Lim et al., 2021) due to resource constraints. A bigger hyperparameter search would improve our results.

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
