# OpenReview forum: "GLINKX: A Scalable Unified Framework For Homophilous and Heterophilous Graphs"
_ICLR.cc/2023/Conference — Submitted to ICLR 2023_

### Official Review · Reviewer_E2RU · 2022-10-25

**Confidence:** 2
**Correctness:** 3
**Technical Novelty And Significance:** 2
**Empirical Novelty And Significance:** Not applicable
**Recommendation:** 3

**Clarity, Quality, Novelty And Reproducibility:**

- The novelty, clarity should be improved.
- The reproducibility is acceptable.

**Strength And Weaknesses:**

Strength
- It is interesting to make LINKX both inductive by using label propagation.

Weaknesses
- This is not the first to consider monophily in GNNs, such as H2GCN and EvenNet, although this may be the first to imcorporate it with label propagation.
- The writing and organization are poor. Many components are redundant, such as the propagation formula in main body, algorithm and figure 1.
- Some parts of the paper is not clear, such as the meaning of Figure 3.
- The experimental evaluations are insufficient. The employed datasets are too little. Besides, the performance improvements is marginal compared with LINKX.

**Summary Of The Paper:**

This paper extends LINKX by considering the monophily with monophilous label propagation. A graph is monophilous if the label of a node is similar to that of its neighbors’ neighbors.  To this end, it proposes two stages label propagations to meet the monophily property. Experimental evaluations show the marginal performance improvement.


**Summary Of The Review:**

My main concerns are the weak novelty, poor writing and insufficient evaluations.

---

> ### Author Response · Authors · 2022-11-18
> **Author response to Reviewer E2RU**
>
> We want to thank Reviewer E2RU for the insightful comments.
>
> **H2GCN + EvenNet:** We agree that H2GCN considers monophily too, and to the best of our knowledge, we could not find a specific statement in the EvenNet paper. However,  there are two significant differences between GLINKX and H2GCN/EvenNet:
>
> 1. Both methods face scalability problems (growth of the receptive field, neighbor sampling) as they are similar to GCNs architecture-wise.
> 2. H2GCN is not better than LINKX (see the experiments on pages 8, 9, and 25 of [1])
> 3. Both papers experiment with much smaller datasets than we do.
>
> We would like to point out that the simplicity of our method should not be taken as a weakness. The monophilous label propagation (MLaP) is very simple, but at the same time, we observe that it leads to improvement in performance. Considering the simplicity and the cost – compared to methods such as H2GCN/EvenNet – we argue that we already have a strong contribution.
>
> **Writing and Organization:** We have reorganized the paper and improved its clarity in the revised PDF (see our general response).
>
> **Meaning of Figure 3:** Figure 3 shows the homophily distribution of one of the datasets (yelp-chi) we run experiments with. We put this figure because we wanted to highlight that real-world networks can have homophilous and heterophilous regions while being monophilous. Spam networks, such as yelp-chi, where reviews are to be classified as spam/non-spam, are typical examples of monophilous networks with homophilous and heterophilous regions. Roughly, spam reviews link to non-spam reviews (creating heterophilous areas), while non-spam reviews link to other non-spam reviews (creating homophilous areas).
>
> **Other contributions:** We disagree that our contribution is just the practical method. In this paper, we present a very simple method (compared to the aforementioned methods), and such a method is theoretically grounded, which none of the existing methods do (to the best of our knowledge). For completeness, we refer the Reviewer to the general comment.
>
> **References**
>
> [1] Lim, Derek, et al. "Large scale learning on non-homophilous graphs: New benchmarks and strong simple methods." Advances in Neural Information Processing Systems 34 (2021): 20887-20902.

---

### Official Review · Reviewer_XQ6r · 2022-10-25

**Confidence:** 4
**Correctness:** 3
**Technical Novelty And Significance:** 2
**Empirical Novelty And Significance:** 3
**Recommendation:** 5

**Clarity, Quality, Novelty And Reproducibility:**

Clarity: This paper is well-organized and easy to follow.

Quality: This paper is of good quality with detailed discussion and comprehensive experiments.

Novelty: Technically, the novelty of this paper is limited because the proposed framework is a combination of existing methods, e.g., KEGs and LP.

Reproducibility: it should be possible to implement the method and reproduce the results from the description given in the paper.

**Strength And Weaknesses:**

Strengths:
- The paper is well-organized and easy to follow and the target problem of GNNs for homophilous and heterophilous graphs is very important.
- The proposed method is effective and efficient in handling homophily and heterophily of large-scale graphs with some theoretical guarantee.
- Extensive experiments have been conducted to demonstrate the superiority of the proposed method from different aspects.

Weakness:
- The novelty of this paper is limited: the proposed framework is the combination of KGEs and LP.
- There are different ways to get positional embeddings, i.e., the embeddings that can preserve global/local structures of graphs. For example, traditional structural role discovery methods such as RolX [1] and network embedding methods such as node2vec [2]. So the questions are:
1) what are the reasons to make use of KGEs as PEs?
2) it is possible to compare different PEs?
- If there are errors/noises from the 1st stage of PEs, is it possible to handle the noises in the following stages?

There are also some minor concerns:
- Page 3: what is the meaning of \epsilon?
- Page 4: circumventing the to do
- Page 5: Alg. 1 (??)

[1] RolX: Structural Role Extraction & Mining in Large Graphs

[2] node2vec: Scalable Feature Learning for Networks


**Summary Of The Paper:**

This paper studies the problem of GNNs under homophily and heterophily and the authors propose a scalable unified framework named GLINKX for this problem. The proposed GLINKX works in three separate stages:
- Making use of KEG to learn positional embeddings.
- Propagating information about labels from node's neighbors.
- Propagating the predicted labels and making final predictions.

GLINKX is also guaranteed with theories. Experimental studies on large-scale homophilous and heterophilous graphs demonstrate the effectiveness and efficiency of the proposed method.

**Summary Of The Review:**

This paper studies an interesting and important problem. Overall this paper is well-organized and easy to read. Experiments from different aspects have been conducted. However, I have some concerns about the novelty and the selection of one component. Therefore, I will recommend a weak rejection of the paper, which I am willing to turn into an acceptance if all issues are addressed.

---

> ### Author Response · Authors · 2022-11-18
> **Author response to Reviewer XQ6r**
>
> We want to thank Reviewer XQ6r for the insightful comments. Regarding the novelty of the contributions and the usage of KGEs, we redirect the Reviewer to our main comment.
>
> **Additional KGE methods:** We agree that one could use other methods of encoding positional information, such as node2vec and RolX, in place of KGEs (however, they will not be able to work on heterogeneous graphs). Our thought behind using KGEs is scalability in training – since tools such as Pytorch-Biggraph – can make this possible in large-scale settings and incorporate heterogeneity in relations (if it exists).
>
> For node2vec, we need to do random walks (which is equivalent to sampling trajectories), which has a higher cost than training KGEs, which only require edges.
>
> **KGE Methods Comparison:** Regarding additional comparisons, our goal through this paper is to highlight different components that can be used to perform node classification in a scalable manner while also maintaining useful properties such as simplicity and inductivity.
>
> It is certainly possible to compare different PEs to measure their performance, and we believe such a comparison constitutes an independent contribution. Hence, we leave this comparison for future work.
>
> For the contributions of our paper, please refer to the general comment.
>
> **Minor comments:**
>
>  * Meaning of $\xi$ -> Thanks for finding the typo; the correct letter is $X$.
>  * Due to space constraints, we have used abbreviations when referencing elements from our text (e.g., Alg. -> Algorithm, Fig. -> Figure, etc.)

---

### Official Review · Reviewer_gBBT · 2022-10-25

**Confidence:** 4
**Correctness:** 2
**Technical Novelty And Significance:** 1
**Empirical Novelty And Significance:** 1
**Recommendation:** 3

**Clarity, Quality, Novelty And Reproducibility:**

The paper writing can be improved significantly.
1. How is a graph treated as a Knowledge Graph? Simply saying that one learns Pytorch Biggraph embedding seems to be pointless. Why does it make sense to treat a general graph as a Knowledge Graph? How does it encode positional information?
2. Many odd typos. For example, Algorithm 1 says Loss is maximized. I am assuming it should be minimized. Also, the notations can be improved to understand what they are. For example, Algorithm 1 says "learn distribution of a node's neighbors", but the y's are still running over the training data. So, it is extremely confusing, where is the neighbor information? What is backward label propagation in stage 3? the expression seems identical to the stage 2.
3. Why does this method capture long range information which is required for heterophily? There seems to some justification in Theorem 2, but the Theorem 2 statement is hard to follow and understand. However, it still seems that the model is utilizing 2 hop information. However, does that mean that the paper is claiming that 2-hop information is sufficient for heterophily? How does one prove this?
4. Comparisons with non-scalable models can still be done for smaller datasets, if only to understand the trade-off between scalability and accuracy.
5. Include training time, memory usage etc. information to show the scalability aspect of the model.

The novelty aspects are difficult to determine and lack of clarity makes the work hard to reproduce.

**Strength And Weaknesses:**

# Strengths:
1. Building scalable shallow method is a very important problem in the area of Graph Neural Networks.
# Weaknesses:
1. The paper and the algorithm are both difficult to follow and understand.
2. Since scalability is the primary goal, some information on training time, memory usage etc. would have been useful.
3. Albeit, the related work cites several papers, but comparison seems to drop them due to the fact that these methods are not scalable. However, there are a few datasets of fairly small size in the experiment, so at least for those, the numbers for these other methods can still be reported, if only to understand whether there is a trade-off involved between accuracy and scalability, and how much is it?

**Summary Of The Paper:**

The paper proposes an extension of LINKX to be robust on both heterophilic and homophilic graphs.

**Summary Of The Review:**

The paper proposes an extension of LINKX model to address both heterophily and homophily. However, the paper requires significant rewrite to clearly highlight how the algorithm works and why should we expect it to work for heterophilic scenario. The experiment should still compare with non-scalable methods on datasets which are small enough. Also, the paper should report training time, memory usage etc. to understand the scalability aspect of the model.

As it stands, I am inclined to reject the paper.

---

> ### Author Response · Authors · 2022-11-18
> **Author response to Reviewer gBBT**
>
> We thank Reviewer gBBT for the insightful comments.
>
> **Clarity** We redirect to the general comment that addresses concerns.
>
> **Comparisons:** We agree that one can always add more experiments to improve the claim. However, we compared our method with sufficient baselines to compare against, and such baselines can be used in a realistic setting. We also provide theoretical justifications for our method.
>
> **Scalability:** We note here that our method performs row-wise mini batching which is substantially less costly and more feasible than neighborhood sampling (which GCNs/GATs do) with the current infrastructure.
>
> In addition to the experiments, we have provided rigorous runtime complexity bounds for the inference time of GLINKX (see Sec. . Training and inference costs – apart from the propagation step that happens once and outside of the training loop – are the same as in the case of LINKX. Also, the training time is affected by the complexity of the model (depth of MLPs, number of hidden channels) we choose, and thus runtime is also a function of how complex the model is.
>
> **Knowledge graph embeddings:** On the graph datasets, we assume that one relation corresponds to all edges (see the general comment). If the graph is heterogeneous, we can consider multiple relations, and the extension of the method to heterogeneous graphs constitutes interesting future work. See our general comment to all Reviewers on why we chose KGEs to encode positional information.
>
> **Additional comments on clarity:**.
>
> 1. **Algorithm 1:** $L_{CE,1}$ and $L_{CE,2}$ correspond to the _negative_ cross-entropy (with a “+” sign) which is maximized. When we learn the distribution of neighbors, for every node, we calculate $\hat y_i = \frac {\sum_{j \in V_{train} : (j, i) \in E_{train}} y_j} {| \{ j \in V_{train}: (j, i) \in E_{train} \}|}$ for all $i \in V_{train}$.
> Then we train a model $\tilde y_i = f_1(x_i, p_i; \theta_1)$ to match $\hat y_i$ by maximizing the negative cross-entropy $L_{CE,1} = \sum_{i \in V_{train}} \sum_{j \in [c]} \hat y_{ic} \log ({\tilde y}_{ic})$ in order to “match” $\hat y_i$. We have clarified this in the revision.
> 2. **Notation of the $y$’s:** There have been some typos in the paper; thank you for identifying them! The correct form of Eq. 3 is $y_i' = \frac {\sum_{j \in V : (i, j) \in E} \tilde y_j} {| \{ j \in V : (i, j) \in E \}|}$ for all $i \in V_{train}$. (there was an arrow in the summation pointing from $j$ to $i$ where it should point from $i$ to $j$, as in the figures). We have made the appropriate corrections in the revision and have improved the notation.
> 3. The expression of Stage 3 (the last step of Stage 2 in the revised version) is a label propagation that is different from the other propagation step in three essential ways:
>    1. We use the soft labels $\tilde y_i$ from the first shallow model
>    2. The directionality of the edges is reversed. Note that in Eq. 1, the direction is from $j$ to $i$ ($j \to i$), and in Eq. 3, the direction is from $i$ to $j$ ($i \to j$). We have clarified this better in the revision.
>
> The above changes have been marked with blue in the revised manuscript.
>
> **Training time, memory usage, non-scalable models:** See our response to Reviewer shIF.
>
> **Theorems:**
>
> * We explain the logic between both of the theorems in Sec 3.4 before stating each of the theorems.
> * Theorem 1 argues that training a parametric model to learn the neighbors’ distribution is (much) better than naively counting the number of neighbors in each class. Please refer to paragraph “Justification of stage 2” in Sec 3.4 for a more detailed explanation.
> * Theorem 2 justifies Stages 2 and 3 of our algorithm, where we prove that using additional information about a node’s neighborhood (in the form of a parametric model) together with a shallow model achieves a smaller error at the end of the training, compared to a shallow model that just takes the features as input (such as LINKX). We want to highlight that this result holds _regardless_ of the graph being homophilous or heterophilous. Please refer to paragraph “Justification of stages 2 and 3” in Sec 3.4 for a more detailed explanation.
>
> **Higher-order information + Heterophily:** In its current form, GLINKX uses information from the immediate neighbors of a node in order to keep things simple. Even with the information we have, we observe that it is comparable with the existing baselines. We also agree that having higher-order information will help, and our proposition is complementary to what existing literature has done for higher-order information, and it can thus be incorporated in our setting (see also our response to Reviewer shIF).

---

### Official Review · Reviewer_shiF · 2022-11-02

**Confidence:** 4
**Correctness:** 3
**Technical Novelty And Significance:** 2
**Empirical Novelty And Significance:** 2
**Recommendation:** 3

**Clarity, Quality, Novelty And Reproducibility:**

Novelty is quite limited. The proposed method is a direct combination of several existing works, e.g., GCN, LINKX, Label propagation.

Clarity needs improvements. The motivations of the paper are not quite clear. Some existing works can already automatically balance the homophily and heterophily while learning for the downstream tasks. Authors need to justify what's new against these existing works.

Writings of this work needs to be polished a lot, addressing the typos, grammar issues, and better presentations.


**Strength And Weaknesses:**

Strength:
- The proposed method is light-weight and thus can be used for large graphs.

Weaknesses:
- The proposed method is not applicable to many graphs as it requires knowledge graph embeddings to provide node positional encodings.
- The novelty of the paper is quite limited. It's a incremental combination of the existing methods.
- Experimental results do not show better performance on homophily graphs.
- On heterophily graphs, there are lots of many other approaches that need to be compared with.

**Summary Of The Paper:**

The paper proposes a method GLINKX that combines knowledge graph embedding, ego embeddings, and label propagation to tackle node classification on both homophily and heterophily graphs. Experiments are conducted on several OGB benchmarks and show some effectiveness.

**Summary Of The Review:**

Regarding the details of some weaknesses above:
- Using knowledge graph embeddings as the pre-trained position encodings does not make sense, especially on the datasets where no knowledge can be used for pre-training, e.g., Cora, Citeseer, etc. There's no relation semantic at all.
- On homophily graphs, the proposed method actually does not perform better even than a two-layer GCN. On heterophily graphs, the proposed method performs better than the baseline methods used in the paper, but there are many other more powerful GNN models in the literature and the authors are suggested to compare with those methods.

---

> ### Author Response · Authors · 2022-11-18
> **Author response to Reviewer shIF**
>
> We would like to thank Reviewer shIF for the insightful comments.
>
> **Use of KGEs & Novelty Aspects:** Please refer to the general comment, which addresses the concerns regarding KGEs and Novelty.
>
> **Clarity:** We have uploaded a new version addressing typos, grammar, and presentation issues.
>
> **Comparison with other heterophilous methods and LINKX as a baseline:** We have not compared with other heterophilous methods for two reasons:
>
> 1. LINKX is a very strong baseline, and we found that it outperforms GNN-based methods (H2GCN/EvenNet/[4]). GNN-based methods are not as scalable as LINKX. Thus we do not compare those methods.
> 2. To the best of our knowledge, based on the design desiderata we required from the methods (and the existence of implementations), the methods that can be compared as baselines with our approach were LINKX and FSGNN.
>
> If there are other recent methods that are better than LINKX, which we may have omitted, we kindly ask the Reviewer to point out the baselines that are better than LINKX and are also not GNN-based.
>
> **Homophily and higher-order GNNs:** Our method is first-order, so the fair GCN/GAT comparison baseline is the one-layer GNN since they do 1-hop propagations. Comparison with higher-order GCN/GAT has been added for completeness purposes because higher-order GCN/GAT (i.e., with at least 2 layers) is very hard to scale (neighbor sampling for mini batching). It is also possible, as future work, to extend our method to attain higher-order information from diffusing the features and/or labels (see [1, 2]), which we conjecture is going to perform better than higher-order GCNs/GATs with the same number of hops. Finally, as we already do in the paper, we want to highlight that our method is complementary and can thus be combined with any of the pre-existing methods, even GCNs.
>
> **References:**
>
> [1] Rossi, Emanuele, et al. "Sign: Scalable inception graph neural networks." arXiv preprint arXiv:2004.11198 7 (2020): 15.
>
> [2] Zhang, Wentao, et al. "Graph attention multi-layer perceptron." arXiv preprint arXiv:2206.04355 (2022).
>
> [4] Di Giovanni, Francesco, et al. "Graph neural networks as gradient flows." arXiv preprint arXiv:2206.10991 (2022).

---

### Author Response · Authors · 2022-11-18
**General Comment [Part 1] - Novelty and Contributions**


We would like to thank the reviewers for their helpful feedback and their insightful comments. We have uploaded a revision addressing the reviewers’ comments and all typos.

We want to re-state our paper’s novel contributions and the logic behind them:

1. We present a simple method whose simplicity should not be considered a weakness.
2. We provide novel theoretical justifications for what the method does, which we believe should be considered in the novelty aspects of our contribution.
3. Our method is complementary to other methods in the literature

Below, we give some detailed explanations about the novel components of GLINKX:

**Monophilous Label Propagation**: We introduce Monophilous Label Propagation (aka MLaP), which is a novel component in our overall method.

For better clarity, in the revised version, we have merged the forward and the backward propagation steps into one stage (2nd stage), which we call the “MLaP stage” (see e.g., Algorithm 1).

MLaP performs two crucial tasks and is further theoretically justified in the main paper.

The first sub-component of MLaP is training a model that can predict a node’s neighborhood label distribution. This is important because it makes GLINKX inductive and learns the types of nodes each node is connected to based on its features and position in the graph. Then the soft labels from this procedure are backpropagated on the neighbor nodes, thus making each node aware of its neighbors. Intuitively, MLaP tries to learn the label distribution of a node’s neighbor’s neighbor.  The final model combines this propagated information with the node features and the PEs for each node to make a final prediction. This inductive bias is more general and is applicable to both homophily and heterophily settings as demonstrated by the experiments.

**Theoretical Bounds:** We have proven theoretical bounds that justify the components of MLaP and the combination of label information and features on the final model. Theorem 1 explains that using a parametric model to learn the neighbor's distribution is better than simple counting. Then, Theorem 2 justifies why using the trained model of Theorem 1 combined with a model that acts on a node’s features has a better error (and thus learns the final distributions better) than a model that is trained using solely the node features as inputs, such as in the case of LINKX.

To the best of our knowledge, similar error bounds for node classification do not exist, and thus we argue that our theoretical contribution constitutes a novel component.

**Knowledge Graph Embeddings (KGEs) as Positional Encodings (PEs):** Knowledge graphs are, in general heterogeneous graphs. However, we can think of a homogeneous graph (like Cora, Squirrel, etc.) as a special case with **one edge type**. KGEs as PEs have not been shown to work well explicitly across homophilous and heterophilous settings, and we believe this is an important observation.

In our case, each node $u$ is associated with a positional embedding $p_u \in \mathbb R^{d_P}$, we have only one relation $\rho \in \mathbb R^{d_P}$, which wlog we can set to be a constant vector. In knowledge graph terminology, for every edge $(u ,v)$, we have the head to be $h = p_u$, the tail to be $t = p_v$, and the relation to be $r = \rho$. In the revised version, we have added an algorithm in Appendix A.4 that explains the KGE generation process in detail.

Also, we have used KGEs as PEs **as a way to embed the graph**. This is done for various reasons: First, existing KGE embeddings are a tried-and-tested method of embedding nodes in the literature and can also be scaled to graphs with billions of nodes [1]. There is software that enables us to do it relatively effortlessly [2]. Moreover, KGEs can encode positional information in a low number of dimensions compared to using exact embeddings. Furthermore, using KGEs has the additional benefit of easily extending GLINKX to heterogeneous graphs, which is helpful in real-world social networks where more than one type of relations exists. Finally, KGEs are generally transductive but can also be made inductive (see [1, 3]).

**Datasets:** In the paper, we have shown results on five datasets that contain both homophilous and heterophilous cases, which we believe are sufficient for experimental evidence. Additionally, we have compared against a lot of baselines already. We have also provided the runtime complexity of our method.

**References:**

[1] El-Kishky, Ahmed, et al. "TwHIN: Embedding the Twitter Heterogeneous Information Network for Personalized Recommendation." KDD 2022.

[2] Lerer, Adam, et al. "Pytorch-biggraph: A large scale graph embedding system." PMLR.

[3] Albooyeh, Marjan, Rishab Goel, and Seyed Mehran Kazemi. "Out-of-sample representation learning for knowledge graphs." Findings of the Association for Computational Linguistics: EMNLP 2020. 2020.

---

### Author Response · Authors · 2022-11-18
**General Comment [Part 2] - Changes in the revised PDF**

In our revision, we have performed the following significant changes (highlighted in magenta):

1. We have improved the presentation of GLINKX’s algorithm steps for clarity. More specifically, we combined the label propagation step (which before was in stages 2 and 3 of the algorithm) into one step, which we call MLaP (monophilous label propagation), which is the 2nd stage of the new revision. The 3rd stage now has only the final model training.
2. We provided more motivation about monophily (See Sec. “Homomphily, Heterophily & Monophily”) and why it is helpful for real-world, large-scale graphs, which usually have both homophilous and heterophilous regions. More specifically, we have provided additional motivation in Figure 3, where we have provided further context on why the concept of monophily is interesting and how it can appear in a real-world network in the context of identifying spam reviews.
3. We have added a section in Appendix A.4 for training KGEs as PEs.
4. We have addressed individual Reviewer comments to improve the clarity of the paper.

---

### Decision · Program_Chairs · 2023-01-20

**Decision:**

Reject

**Justification For Why Not Higher Score:**

The major issues were those related to comparison with baselines (as mentioned in my review), particularly w.r.t. heterophily GNNs, higher order GNNs, and the lack of computation time / memory results.

**Justification For Why Not Lower Score:**

N/A

**Metareview: Summary, Strengths And Weaknesses:**

The paper proposes a method GLINKX that uses (1) knowledge graph embeddings (representing graph positional information), (2) node features, and (3) a novel Monophilous Label Propagation (MLaP) approach utilizing node labels. They also provide theoretical justification, which justify the benefits of their design based on error bounds (particularly the parametric model in MLaP, and the two-stage design).

In general, reviewers find the problem interesting and important, and appreciate the novel theoretical grounding in the form of error bounds. However, there are a number of key issues which multiple reviewers agreed upon:

**Comparison with baselines:**
- Reviewers noted that the method lacks comparison with GNNs designed for the heterophily setting. In the author response, they state that one reason is that GNN based methods are not as scalable as LINKX. However, this would be more convincing if performance results on the time / memory consumption in practice were provided.
- Reviewers also mentioned that the method also does not clearly outperform higher order GCNs (with >1 layer). The authors state in their response that the fair comparison is to 1-layer GCNs, since their method is also 1 layer, but without results showing that GLINKX can also improve its performance with higher number of layers, or performance results showing that its time / memory consumption compares favorably, this is still not fully convincing.

**Clarity:**
- Reviewers pointed to a number of issues related to clarity. Many of these have been resolved, and I thank the authors for their efforts in improving the clarity. However, I believe that the paper would still benefit from making the idea of using knowledge graph embeddings clearer and more self-contained, particularly for readers who are not already familiar with such methods

In the end, reviewers and AC agree that while the work has intriguing ideas, due to the above issues, the work is not yet ready for publication at ICLR. The reviews offer a number of helpful suggestions for improvement, so I encourage the authors to continue improving the paper based on the reviews for future submissions.